# Contrastive Reinforcement Learning of Symbolic Reasoning Domains

**Gabriel Poesia**
Stanford University
poesia@cs.stanford.edu

**WenXin Dong**
Stanford University
wxd@stanford.edu

**Noah Goodman**
Stanford University
ngoodman@stanford.edu

## Abstract

Abstract symbolic reasoning, as required in domains such as mathematics and logic, is a key component of human intelligence. Solvers for these domains have important applications, especially to computer-assisted education. But learning to solve symbolic problems is challenging for machine learning algorithms. Existing models either learn from human solutions or use hand-engineered features, making them expensive to apply in new domains. In this paper, we instead consider symbolic domains as simple environments where states and actions are given as unstructured text, and binary rewards indicate whether a problem is solved. This flexible setup makes it easy to specify new domains, but search and planning become challenging. We introduce five environments inspired by the Mathematics Common Core Curriculum, and observe that existing Reinforcement Learning baselines perform poorly. We then present a novel learning algorithm, Contrastive Policy Learning (ConPoLe) that explicitly optimizes the InfoNCE loss, which lower bounds the mutual information between the current state and next states that continue on a path to the solution. ConPoLe successfully solves all four domains. Moreover, problem representations learned by ConPoLe enable accurate prediction of the categories of problems in a real mathematics curriculum. Our results suggest new directions for reinforcement learning in symbolic domains, as well as applications to mathematics education.

## 1 Introduction

Humans posses the remarkable ability to learn how to reason in symbolic domains, such as arithmetic, algebra, and formal logic. Our aptitude for mathematical cognition builds on specialized neural bases but extends them radically through formal education [6, 16, 11, 12]. Learning to reason in symbolic domains poses an important challenge for artificial intelligence research. As we describe below, this type of reasoning has unique features that distinguish it from domains in which machine learning has had recent success.

From a practical viewpoint, since symbolic reasoning skills span years of instruction in school, advances in symbolic reasoning may have a large impact on education. In particular, automated tutors equipped with step-by-step solvers can provide personalized help for students working through problems [28], and aid educators in curriculum and course design by semantically relating exercises based on their solutions [22, 23]. Indeed, studies have found automated tutors capable of yielding similar [3, 20] or larger [29] educational gains than human tutors. While *solving* problems alone does not necessarily translate to good *teaching*, automated tutors typically have powerful domain models as their underlying foundation.

However, even modest mathematical domains are challenging to solve. As an example, consider solving linear equations step-by-step using low-level axioms, such as associativity, reflexivity and operations with constants. This formulation allows all solution strategies that humans employ to be

expressed as combinations of few simple rules, making it attractive for automated tutors [28, 25]. But while formulating the domain is simple, obtaining a general solver is not. Naïve search is infeasible due to the combinatorial solution space. As an example, the search-based solver used in the recent Algebra Notepad tutor [25] is limited to solutions of up to 4 steps. An alternative is manually writing expert solver heuristics. Again, even for a domain such as high-school algebra, this route is difficult and error-prone. As we describe in Section 5.2, we evaluated Google MathSteps, a library that backs educational applications with a step-by-step algebra solver, on the equations from the Cognitive Tutor Algebra [28] dataset. MathSteps only succeeded in 76% of the test set, revealing several edge cases in its solution strategies. Thus, even very complex expert-written strategies may have surprising gaps.

An alternative could be to *learn* solution strategies via Reinforcement Learning (RL). We formulate symbolic reasoning as an RL problem of deterministic environments that execute domain rules and give a binary reward when a problem is solved. Since we aim for generality, we assume a domain-agnostic interface with the environment: states and actions are given to agents as unstructured text. These domains have several idiosyncrasies that make them challenging for RL. First, trajectories are unbounded, since axioms might always be applicable and lead to new states (e.g. adding a constant to both sides of an equation). Second, agents have no direct access to the underlying structure of the domain, only observing strings and sparse binary rewards. Finally, each problem only has one success state (e.g. `x = number`, in equations). These properties rule out many popular algorithms for RL. For instance, Monte Carlo Tree Search (MCTS, [7]) uses random policy rollouts to train its value estimates. If the solution state is unique, such rollouts only find non-zero reward if they happen to find the complete solution. Thus, MCTS fails to guide search toward solutions [1]. Indeed, as we show in Section 5.2, Deep Q-Learning, and other algorithms that are based on estimating expected rewards, perform poorly in these symbolic domains.

To overcome these challenges, we propose a novel learning algorithm, Contrastive Policy Learning (ConPoLe), which succeeds in symbolic environments. Our key insight is to directly learn a policy by attempting to capture the mutual information between current and future states that occur in successful trajectories. ConPoLe uses iterative deepening and beam search to find successful and failed trajectories during training. It then uses these positive and negative examples to optimize the InfoNCE loss [24], which lower bounds the mutual information between the current state and successful successors. This provides a new connection between policy learning and unsupervised contrastive learning. Our main contributions in this paper are:

- We introduce 5 environments for symbolic reasoning (Fig. 1) drawn from skills listed in the Mathematics Common Core Curriculum (Section 3). We find that existing Reinforcement Learning algorithms fail to solve these domains.

- We formulate policy learning in deterministic environments as contrastive learning, allowing us to sidestep value estimation (Section 4). The algorithm we introduce, ConPoLe, succeeds in all five Common Core environments, as well as in solving the Rubik's Cube (Section 5.2).

- We provide quantitative and qualitative evidence that the problem representations learned by ConPoLe reflect the equation-solving curriculum from the Khan Academy platform. This result suggests a number of applications of representation learning in education.

## 2 Related Work

Automated mathematical reasoning spans several research communities. Theorem-proving languages, such as Coq [5] or Lean [10], enable the formalization of mathematical statements and can verify proofs, but they are limited in their ability to *discover* solutions automatically. A rich line of recent work has focused on learning to produce formal proofs using supervised [27, 32, 2] and reinforcement learning [4, 18, 26]. Similar to GPT-f [27], our model only assumes unstructured strings as input; however, we do not use a dataset of human solutions. To the best of our knowledge, our work is the first to consider learning formal reasoning directly from text input (like GPT-f) by purely interacting with an environment (like rlCop [18] and ASTatic [32]).

Existing RL algorithms for theorem proving make use of partial rewards to guide search. For example, rlCop [18] learns to prove in the Mizar Verifier using Monte Carlo Tree Search. In Mizar, applying a theorem decomposes the proof into sub-goals. Random policy rollouts often close a fraction of the sub-goals, providing signal to MCTS. In our environments, however, there is only one solution state

**sorting** [====|==|=|===] → [===|=|==|====] → [=|===|==|====] → **[=|==|===|====]**

**ternary-addition** #(b2 c2 c0 c2) → #(a2 b3 c0 c2) → #(b3 c0 c2) → #(c0 b3 c2) → **#(c0 c2 b3)**

**multiplication** 86e3 * 28e2      (86e3 * 2e3) + (86e3 * 8e2)      (172e6 + 688e5)
     (86e3 * (2e3 + 8e2)) ↗    (172e6 + (86e3 * 8e2)) ↗ ↘ **240800000**

**fractions** 1/6 + (1*2)/(3*2)      1/6 + 2/6      3/6    **1/2**
1/6 + 1/3 ↗    → 1/6 + 2/(3*2) ↗    ↘ (1+2)/6 ↗    ↘ 3/(2*3) ↗

**equations** 2x + 1 = 5    2x + (1 - 1) = 4 → 2x + 0 = 4    (x*2)/2 = 4/2 → x*(2/2) = 4/2    **x = 2**
(2x + 1) - 1 = 5 - 1 → (2x + 1) - 1 = 4    2x = 4 → (2x)/2 = 4/2    x*1 = 4/2 → x = 4/2

Figure 1: Example of one problem and step-by-step solution in each CommonCore environment. In `equations`, a problem is a linear equation with the four basic operations and arbitrary parentheses, and actions are applications of low-level axioms (e.g. commutativity, associativity, distributivity, calculations with constants and applying operations on both sides). `ternary-addition` simulates step-by-step arithmetic with carry in base 3: inputs are sequence of $(digit, power)$ pairs, where letters are digits ($\mathtt{a} = 0$, $\mathtt{b} = 1$, $\mathtt{c} = 2$) and numbers are powers of 3 (so $\mathtt{b2} = 1 \cdot 3^2$). Adjacent digits can be added together, and digits 0 can be eliminated. In the example, we start with $1\cdot3^2 + 2\cdot3^2 + 2\cdot3^0 + 2\cdot3^2$ and obtain $2 \cdot 3^0 + 1 \cdot 3^1 + 2 \cdot 3^2$. In `multiplication`, addition and single-digit multiplication are primitive operations, and must be combined with axioms that split larger numbers to perform long multiplication. In `fractions`, operations include factoring numbers into a prime multiplied by a divisor, canceling common factors and combining fractions with the same denominator. Finally, in `sorting`, the agent needs to sort delimited substrings by length by either reversing the entire list or performing adjacent swaps.

for each problem, causing MCTS to degenerate to breadth-first search. This challenge is similar in other large search problems, such as the Rubik's Cube. DeepCubeA [1] handles the sparsity problem in the Rubik's Cube by generating examples in reverse, starting from the solution state. This uses the fact that moves in the Rubik's Cube have inverses, and that the solution state is always known a priori. In contrast, the algorithm we propose in Section 4, which has neither of these assumptions, is still able to learn an effective Rubik's Cube solver; we compare it to DeepCubeA in Section 5.2.

Intelligent Tutoring Systems [20, 28, 22] provide a key application for step-by-step solvers. Tutors for symbolic domains, such as algebra [28, 25] or database querying [3], can use a solver to help students even in the absence of expert human hints (unlike *example-tracing tutors*, which simply follow extensive problem-specific annotations [30]). Currently, solvers used in automated tutors are either search-based [25], but limited to short solutions, or hand-engineered [17]. Our method can learn high-level solution strategies by composing simple axioms, generalizing to long solutions without the need for expert heuristics. Moreover, our analysis in Section 4 suggests that learned problem embeddings can be applied in mathematics education – another advantage of a neural solver.

Finally, our approach to Reinforcement Learning in symbolic domains builds on unsupervised contrastive learning, specifically InfoNCE [24]. Contrastive learning has been used to learn representations that are then used by existing Deep RL algorithms [24, 19, 13]. We instead make a novel connection, casting policy learning as contrasting examples of successful and failed trajectories.

## 3 Setup

Motivated by mathematics education, we are interested in formal domains where problems are solved in a step-by-step fashion, such as equations or fraction simplification. While various algorithms could compute the final solution to such problems, finding a *step-by-step* solution by composing low-level axioms is a planning problem. Formally, we define a *domain* $\mathcal{D} = (S, A, T, R)$ as a deterministic Markov Decision Process (MDP), where $S$ is a set of states, $A(s)$ is the set of actions available at state $s$, $T(s_t, a)$ is the transition function, deterministically mapping state $s_t \in S$ and action $a \in A(s_t)$ to the next state $s_{t+1}$, and $R(s)$ is the binary reward of a state: $R(s) = 1$ if $s$ is a "solved" state, otherwise $R(s) = 0$. States with positive reward are terminal. Initial states can be sampled from a probability distribution $P_I(s)$, which we assume to have states of varying distance to the solution

Table 1: Common Core environments for symbolic reasoning.

| Environment | Reference | Average Branching Factor | BFS Success Rate |
|---|---|---|---|
| sorting | CCSS.Math.Content.MD.A
Sort objects by length. | 5.84 | 17% |
| ternary-addition | CCSS.Math.Content.1.NBT.C
Perform step-by-step arithmetic with carry. | 10.55 | 9% |
| multiplication | CCS.Math.Content.1.NBT.C
Multiply multi-digit whole numbers | 8.62 | 26% |
| fractions | CCSS.Math.Content.NF.B
Manipulate expressions with fractions. | 6.58 | 18% |
| equations | CCSS.Math.Content.8.EE.C
Solve linear equations in one variable. | 27.12 | 5.5% |

state. This implicit curriculum allows agents to find a few solutions from the beginning of training – an important starting signal, since we assume no partial rewards.

States in symbolic domains typically have a well-defined underlying structure. For example, mathematical expressions, such as equations, have a recursive tree structure, as do SQL queries and programming languages. A different, yet similarly well-defined structure dictates valid states in the Rubik's cube. However, in order to study the general problem of symbolic learning, we want to assume no structure for states beyond the MDP specification. Therefore, we assume all states and actions are strings, i.e. $S, A(s) \subseteq \Sigma^*$ for some alphabet $\Sigma$. Naturally, our goal is to learn a policy $\pi(a|s)$ that maximizes the probability that following $\pi$ when starting at a randomly drawn state $s_0 \sim P_I(\cdot)$ leads to a solved state $s_k$.

## 3.1 Environments

We introduce four environments that exercise core abilities listed in the Mathematics curriculum of the Common Core State Standards. The Common Core is an initiative that aims to build a standard high-quality mathematics curriculum for schools in the United States, where "forty-one states, the District of Columbia, four territories, and the Department of Defense Education Activity (DoDEA) have voluntarily adopted and are moving forward with the Common Core"[1]. We draw inspiration from four key contents in the curriculum: "Expressions and Equations", "Numbers and Operations – Fractions", "Measurements and Data", and "Operations and Algebraic Thinking".

Table 1 lists these environments. To put their respective search problems into perspective, we report two statistics: the average branching factor of $1M$ sampled states, and the success rate of a simple breadth-first search (BFS) in solving 1000 sampled problems, when limited to visiting $10^7$ state-action edges. All the environments require strategy and exploiting domain structure to be solved: naïve search succeeds only on the simplest problems, with success rates ranging from $26\%$ in `multiplication` to $5.5\%$ in `equations`. We describe all axioms available in each environment in detail in the Appendix. Figure 1 gives an example of a problem and step-by-step solution in each of the environments. Agents operate in a fairly low-level axiomatization of each domain: simple actions need to be carefully applied in sequence to perform desirable high-level operations. For example, to eliminate an additive term in one side of an equation, one needs to subtract that term from both sides, rearrange terms using commutativity and associativity to obtain a sub-expression with the term minus itself, and finally apply a cancelation axiom that produces zero. This formulation allows agents to compose actions to form general strategies, but at the same time makes planning challenging.

## 4 Contrastive Policy Learning

A common paradigm in Reinforcement Learning uses rollouts to learn value estimates: a starting state is sampled, the agent executes its current policy until the trajectory horizon, and state value is updated using observed rewards. However, when rewards are sparse, as in the symbolic environments

---

[1] http://www.corestandards.org/about-the-standards/

considered here, this paradigm suffers from a serious *attribution problem*. In failed trajectories, which are typically the vast majority encountered by an untrained agent, the signal of a null reward is weak: failure could have happened due to any of the steps taken. Moreover, the agent has no direct feedback on what *would have happened* had it taken a different action in a certain state, until it visits another similar state. In turn, long intervals between similar observations can hinder leaning. These issues are ameliorated by Monte Carlo Tree Search, and similar algorithms, which explore a search tree and may abandon a sub-tree based on expected outcomes. However, MCTS relies on Monte Carlo estimates of the value of candidate states. In games, such as Go or Chess, random rollouts find a terminal state per problem, providing reward signal. (Indeed, self-play in such games always terminates and wins about half of games.) But in domains with a single terminal state, like math problems or the Rubik's cube, such estimates are unhelpful [1], returning zero reward with probability that approaches 1 exponentially in the distance to the solution.

To obtain signal about each step of a solution, we would like to consider multiple alternative paths, as MCTS does, without relying on random rollouts to yield useful value estimates. To that end, we employ *beam search* using the current policy $\pi$, maintaining a beam of the $k$ states most likely to lead to a solution. When a solution is found, the beam at each step contains multiple alternatives to the action that eventually led to a solution. We take those alternatives as *negative examples*, and train a policy that maximizes the probability of picking the successful decision over alternatives.

Let $f_\theta(p, s_t)$ be a parametric function that assigns a non-negative score between a proposed next state $p$ and the current state $s_t$. Although any non-negative function suffices, here we use a simple log-bilinear model that combines the embeddings of the current and proposed states, both obtained with the state embedding function $\phi_\theta : S \to \mathbb{R}^n$:

$$f_\theta(p, s_t) = \exp\left(\phi_\theta(p)^\top \cdot W_\theta \cdot \phi_\theta(s_t)\right)$$

Once we find a solution using beam search, for each intermediate step $t$ we have a state $s_t$ and a set $X = \{p_1, \cdots, p_N\}$ of next state proposals, obtained from actions that were available at states in the beam at step $t$. One of these, which we call $s_{t+1}$, is the proposal in the path that led to the found solution. We then minimize:

$$\mathcal{L}_t(\theta) = \mathbb{E}_{s_t} \left[ -\log \frac{f_\theta(s_{t+1}, s_t)}{\sum_{p_i \in X} f_\theta(p_i, s_t)} \right]$$

Algorithm 1 describes our method, which we call Contrastive Policy Learning (ConPoLe). In successful trajectories, ConPoLe adds positive and negative examples of actions to a *contrastive experience replay buffer*. After each problem, it samples a batch from the buffer and optimizes $\mathcal{L}_t$. Since we can only find positives when a solution is found, ConPoLe essentially ignores unsolved problems. Moreover, to improve data efficiency, ConPoLe iteratively deepens its beam search: with an uninitialized policy, it cannot expect to find long solutions, so exploring deep search trees is unhelpful. In our implementation, we simply increase the maximum search depth by 1 every $K$ problems solved, up to a fixed maximum depth.

Our loss $\mathcal{L}_t$ is equivalent to the InfoNCE loss, which is shown in [24] to bound the mutual information: $I(s_{t+1}, s_t) \geq \log |X| - \mathcal{L}_t$ . Note that in our domains each state is the next step for the correct solution to *some* problem, thus the negative set found by beam search approximates negative samples of next states from other solutions, though with a bias toward closer 'hard negatives'[31]. Thus our approach can be interpreted as learning a representation that captures the mutual information between current states and their successors along successful solutions. The MI bound becomes tighter with more negatives: we explore this property experimentally in Section 5.2. We refer to [24] for a detailed derivation of this bound and its properties.

As our policy we directly use the similarity function $f(p_i, s_t)$ normalized over possible next states. The objective $\mathcal{L}_t$ is also the categorical cross entropy between the model-estimated next state distribution $\pi_\theta(p|s_t) \frac{f(p,s_t)}{\sum_{p_j \in A(s)} f(p_j, x_t)}$ and the distribution of successful proposals. It is thus minimized when the predictions match $P(s_{t+1} = p_i|s_t)$, the optimal policy.

The ConPoLe approach avoids value estimation by focusing directly on statistical properties of solutions. The `sorting` environment illustrates the intuition that this is a simpler objective. In this

domain, the agent has to sort a list of substrings by their length, using adjacent swaps or reversing the whole list. The value of a state would be proportional to the number of inversions (i.e. pairs of indices that are in the incorrect order). This prediction problem is hard for learning: $O(n^2)$ pairs of indices need to be considered[2]. ConPoLe, however, only needs to tell if a proposed state has more or less inversions than the current state. In the case of adjacent swaps, this amounts to detecting whether the swapped elements were previously in the wrong order. By having negative examples to contrast with successful trajectories, ConPoLe can learn a policy by completely sidestepping value estimation.

---

**Algorithm 1:** Contrastive Policy Learning (ConPoLe)

---

**Input:** Environment $E$
**Output:** Learned policy parameters $\pi_\theta$
$\theta \leftarrow$ `init_parameters()`
$\mathcal{D} \leftarrow \varnothing$
$n\_solved \leftarrow 0$
**for** $episode \leftarrow 1$ **to** $N$ **do**
    $p \leftarrow E.$`sample_problem()`
    $(solution, visited\_states) \leftarrow$ `beam_search`$(E, \pi_\theta, p, beam\_size, max\_depth)$
    **if** $solution \neq$ ***null*** **then**
        $n\_solved \leftarrow n\_solved + 1$
        $neg\_states \leftarrow visited\_states \setminus solution$
        **for** $i \leftarrow 1$ **to** $length(solution) - 1$ **do**
            $pos \leftarrow (solution[i], solution[i+1])$
            $neg \leftarrow \{(solution[i], c) : c \in neg\_states$ `from step i of beam_search`$\}$
            $\mathcal{D}.$`add`$(\langle pos, neg \rangle)$
        **end**
        $B \leftarrow \mathcal{D}.$`sample_batch()`
        $\theta \leftarrow \theta - \alpha \nabla$`InfoNCE`$(\theta, B)$
    **end**
**end**

---

# 5 Experiments

We now evaluate our method guided by the following research questions: How does ConPoLe perform when solving symbolic reasoning problems from educational domains? How do negative examples affect ConPoLe's performance? Are ConPoLe's problem embeddings useful for downstream educational applications? Can ConPoLe be applied to other large-scale symbolic problems?

## 5.1 Setup

We compare ConPoLe against four RL baselines and the Google MathSteps[3] library, which contains manually implemented step-by-step solvers for the Fractions and Equations domains (as opposed to simply giving the final answer, as several other libraries do). The first learning-based baseline is the *Deep Reinforcement Relevance Network* [14, DRRN], an adaptation of Deep Q-Learning for environments with textual state and dynamic action spaces. We additionally test Autodidatic Iteration [21, ADI] and Deep Approximate Value Iteration [1, DAVI] – both methods have been recently used to solve the Rubik's Cube, a discrete puzzle that is similar to the Common Core domains in that it only has a single solution state. Finally, we use a simple Behavioral Cloning (BC) baseline, as done in [8]: it executes a random policy until its budget is exhausted; then, it trains a classifier only on the successful trajectories that picks the successful action.

As a simpler alternative to ConPoLe, we use a baseline we call *Contrastive Value Iteration* (CVI). This algorithm is identical to ConPoLe except for its loss: we train it to predict the final reward obtained by each explored state. In other words, after finding a solution, CVI will add examples to its replay buffer of the form $(s, r) \in \mathcal{S} \times \mathbb{R}$, where $s$ is a visited state and $r$ is 1 if that state occurred in

---

[2]A $\Theta(n \log n)$ solution counts inversions by exploiting the merge sort recursion [9].
[3]`https://github.com/google/mathsteps`

| Agent | Sorting | Addition | Multiplication | Fractions | Equations |
|-------|---------|----------|----------------|-----------|-----------|
| BC | 92.0% | 64.5% | 10.0% | 52.5% | 4.5% |
| DRRN | 29.5% | 40.0% | 10.5% | 20.0% | 2.5% |
| ADI | 91.0% | 63.5% | 9.5% | 46.5% | 5.5% |
| DAVI | 99.5% | 54.0% | 18.5% | 57.5% | 8.0% |
| *MathSteps*$^{(N)}$ | - | - | - | 100% | 76.0% |
| CVI | 77.0% | 72.0% | 100.0% | 84.5% | 89.0% |
| ConPoLe-local | 100.0% | 98.5% | 99.5% | 86.0% | 76.5% |
| ConPoLe | 100.0% | 100.0% | 99.5% | 96.0% | 92.5% |

Table 2: Success rate of all agents in the CommonCore environments. Agents were ran with 3 random seeds for $10^7$ environment steps, and tested every $100k$ steps on a held-out set of 200 problems. We report the best observed success rate of each agent's greedy policy (i.e. no search is done at test time). $(N)$ emphasizes that the MathSteps library is not learning-based: we simply ran it on test problems.

the path to the solution, or 0 otherwise. This can be seen as estimating $P(s')$: the probability that $s'$ would be observed in a successful trajectory, without conditioning on the current state. Since the reward is binary, this corresponds to value estimation.

All models use two-layer character-level LSTM [15] network to encode inputs (DRRN uses two such networks). Each agent was trained for $10^7$ steps in each environment; runs took from 24 to 36 hours on a single NVIDIA Titan Xp GPU. Problems in the `equations` domain come from a set of 290 syntactic equation templates (with placeholders for constants, which we sample between -10 and 10.) extracted from the Cognitive Tutor Algebra [28] dataset. Other environments use generators we describe in the Appendix. Code for agents and CommonCore environments is available at `https://github.com/gpoesia/socratic-tutor`. Our Common Core environments are implemented in Rust, and a simple high-throughput API is available for Python.

## 5.2 Solving CommonCore domains

We start by comparing agents when learning in the CommonCore environments. In this experiment, agents train using sequentially sampled random problems. We define one *environment step* as a query in which the agent specifies a state, and the environment either returns that the problem is solved or lists available actions and corresponding next states. Since trajectories can be potentially infinite, we limit agents to search for a maximum depth of 30, which was enough for us to manually solve a sample set of random training problems from all environments.

Table 2 shows the best performance of each agent on a held-out test set of 200 problems. Success rate is measured using each agent's greedy policy: we do not perform any search at test-time. DRRN fails to effectively solve any of the environments. ADI, DAVI and BC can virtually solve the sorting domain, but this performance does not translate to harder domains multiplication and equations. ConPoLe shows strong performance on all domains, with CVI being comparable in equations and multiplication, but falling behind in the other two domains.

DRRN quickly converges to predicting near 0 for most state-action pairs, failing to learn from sparse rewards. We note that the environments where DRRN has shown success, such as text-based games, have shorter states and actions as well as intrinsic correlations between states (derived from natural language). These features may help to smooth the value estimation problem but are not available in the Common Core domains.

We note that ADI and DAVI have been previously successful in puzzles with an important feature: the state sampler produces problems that are exactly $k$ steps from the solution for all $k$ up to the maximum necessary. For instance, in the Rubik's Cube, scrambled cubes can be generated by starting from the solution state and performing $k$ moves in reverse. This feature is present in the `sorting` domain (in which both ADI and DAVI perform well), but not in general. For example, even the simplest equation in the Cognitive Tutor dataset still requires 3 actions to be solved; some require up to 30, and there are many gaps in this progression. These gaps cause ADI and DAVI to find states at test time that are out of their training distribution. As suggested in [21], we experimented with

replacing the 1-step lookahead of these algorithms by a bounded BFS. However, we found that this naïve exploration strategy is sample inefficient and did not significantly change their performance.

On the other hand, algorithms that learn from contrasting examples (CVI and ConPoLe) perform well in all environments. Using only examples derived from solved problems gives much higher signal to each data point: we only consider a "negative" when we have a corresponding positive example which actually led to a solution. Moreover, we observe gains in using the InfoNCE loss for training with contrasting examples: ConPoLe performs consistently better than CVI, and can very quickly learn to solve almost all sorting and addition problems. In these domains, deciding whether an action moves towards the goal is an easy problem, but precisely estimating values is challenging. In Sorting, for example, CVI's learned policy becomes unreliable for lists with more than 7 elements. We observe a similar limitation in Addition, where one sub-problem is sorting digits by the power of 10 they multiply.

These observations provide evidence to answer our first research question: using contrastive examples is beneficial to learning policies in symbolic domains, and explicitly optimizing a contrastive loss improves results further.

**The impact of negative examples**    Contrastive learning algorithms have been observed to perform better with more negative samples: the variance of the InfoNCE estimator decreases with more negative examples. However the choice of negative examples can also impact the performance [31]. We thus experimented with a simple variant of ConPoLe that produces fewer, but more local, negative examples. Instead of using *all* candidates visited by beam search, we instead only use those that were actual candidates for successors of states in the solution path. We call this variant *ConPoLe-local*, since it only uses "local" negatives. The performance of ConPoLe-local is shown in Table 2. ConPoLe-local behaves like ConPoLe in Sorting, Addition and Multiplication. In the last two domains, however, there is a significant performance gap between ConPoLe and ConPoLe-local. Interestingly, this does not seem to be the case for CVI. We executed the same experiment with CVI, and observed no reliable difference in performance or learning behavior. This finding supports the importance of the connection between reinforcement learning and contrastive learning: using all available negatives, in the way suggested by InfoNCE, yields the best results.

## 5.3    Comparing learned representations to human curricula

A learned neural solver yields distributed representations for problems. One natural question is whether these representations capture properties of the problems that resemble how humans structure the same domains.

To investigate this question, we collected a dataset of equations from the Khan Academy[4] educational portal. We used the equations listed as examples and test problems in the sections of the algebra curriculum dedicated to linear equations. There are four sections: "One-step addition and subtraction equations", "One-step multiplication and division equations", "Two-step equations" and "Equations with variables on both sides". The first three categories had 11 equations, and the last had 9.

We evaluated a 1-nearest neighbor classifier that predicts the equation's category from the closest labeled example, based on four distance metrics: string edit distance (a purely syntactical baseline), and cosine similarity using the representations learned by agents. Chance performance in this task is 26%.

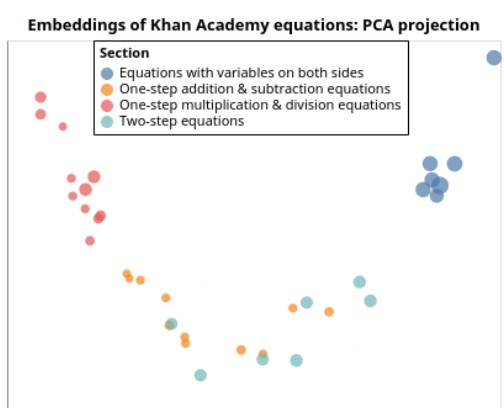

Figure 2: PCA projection of ConPoLe's learned representations for equations from Khan Academy. Point sizes indicate ConPoLe's solution length (ranging from 7 to 29 in these exercises). We observe clusters that closely match the different sections of the Khan Academy curriculum.

---

[4]https://www.khanacademy.org/

Table 3: Accuracy of predicting equation categories from learned representations on Khan Academy.

| Representation | BC | DRRN | ADI | DAVI | CVI | Edit Distance | ConPoLe |
|---|---|---|---|---|---|---|---|
| Accuracy | 0.619 | 0.642 | 0.619 | 0.619 | 0.642 | 0.714 | **0.905** |

Table 3 shows ConPoLe's representations yield accurate category predictions (90.5%), while other representations are less predictive than the Edit Distance baseline. We further observe that ConPoLe's latent space is highly structured (Figure 2): sections from Khan Academy form visible clusters. This happens despite of ConPoLe being trained without an explicit curriculum or examples of human solutions.

### 5.4 Solving the Rubik's Cube

Finally, we apply ConPoLe to a challenging search problem: the Rubik's Cube [1]. This traditional puzzle consists of a 3x3x3 cube, with 12 possible moves that rotate one of the 6 faces either clockwise or counterclockwise. Initially, all 9 squares in each face have the same color. The cube can be scrambled by applying face rotations, and the goal is to bring it back to its starting state. There are $4.3 \times 10^{19}$ valid configurations of the cube, and a single solution state. We compare ConPoLe to DeepCubeA [1], a state-of-the-art solver that learns with Deep Approximate Value Iteration.

For this task, we simply represent the cube as a string of 54 digits from 0 to 5, representing 6 colors, with a separators between faces. We run the exact same architecture we used in the Common Core domains. We train ConPoLe for $10^7$ steps on a single GPU, with a training beam size of 1000, on cubes scrambled with up to 20 random moves. (DeepCubeA observed 10 billion cubes during training time, compared to 10 million environment steps taken by ConPoLe during training.) In test time, following DeepCubeA, we employ Batch Weighted A* Search (BWAS), using our model's predicted log-probabilities as weight estimates.

We find that ConPoLe is able to learn an effective solver. We tested on 100 instances scrambled with 1000 random moves, as used in DeepCubeA's evaluation. ConPoLe succeeds in all cubes (as does DeepCubeA). In finding solutions, BWAS paired with ConPoLe visits an average of 3 million nodes, compared to 8.3 million with DeepCubeA. DeepCubeA solutions however are shorter (21.6 moves on average, compared to ConPoLe's 39.4).[5] Overall, this result validates the generality and promise of ConPoLe for solving challenging symbolic domains.

## 6 Conclusion

We introduced four environments inspired by mathematics education, in which Reinforcement Learning is challenging. Our algorithm, based on optimizing a contrastive loss, demonstrated significant performance improvements over baselines. While we used educational domains as a test bed, our method can in principle be applied to any discrete planning domain with binary rewards. One requirement is that an untrained agent must find enough solutions to assemble initial contrastive examples. Procedural mathematical domains are a natural instance of these. However, our educational environments assume a linear structure, where applying an axiom directly leads to the next state. This assumption breaks in more expressive formulations of formal mathematics (such as first-order logic or dependent type theories), where proofs have a tree structure. Other domains, such as programs, might pose additional challenges because an unbounded number of actions can be available at a state. Adapting ConPoLe for these settings is non-trivial, and poses important challenges for future work.

Our learned solvers can simplify the process of building Intelligent Tutoring Systems. These systems can free educators to focus on more conceptual problems. On the down side, solvers can provide unfair resources for homework and unequal access could exacerbate inequality. Beyond tutoring, we observed that the representations learned by our agents capture semantic properties about problems. This opens up an avenue for additional research on deep representations for educational applications.

---

[5]BWAS can be tuned to trade off shorter solutions for exploring more nodes; we only tested the default parameters in the DeepCubeA implementation. Given DeepCubeA was trained on 1000x more cubes, we would expect its solutions to still be shorter if we matched both algorithms in number of visited nodes.

## Acknowledgements

We thank the anonymous NeurIPS reviewers for the valuable discussion, which significantly improved our work. This work was supported by a NSF Expeditions Grant, Award Number (FAIN) 1918771.

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
