# A  Common Core environments

In Section 3, we briefly described four Common Core-inspired environments: `equations`, `fractions`, `ternary-addition` and `sorting`. We now provide a detailed description of the states, actions and problem generators for each of these environment.

## A.1  `equations`

The `equations` environment exercises the ability to coordinate primitive algebraic manipulations in order to solve an equation. Each problem is a linear equation on a single variable $x$, and actions are valid manipulations of the equation, following simple axiomatic rules. A valid state in this domain is an *equality*, which is comprised of two *expressions*: one on the left and one on the right. In turn, an expression can be one of the following:

**Constant:**  An integer $n$, or a rational $\frac{a}{b}$ with $b \neq 0$,

**Binary operation:**  A (recursively defined) left-hand side expression $e_l$, an operator $op \in \{+, -, \times, /\}$, and a right-hand side expression $e_r$,

**Unary operation:**  The operator $-$ followed by an expression $e_r$,

**Unknown:**  The unknown $x$.

A state is solved only when it is in the form $x = n$, where $n$ is a constant. When representing states as strings, we use the standard mathematical notation, with the detail that we parenthesize all binary operations so that operator precedence is made explicit.

To generate problems in this domain, we leverage the Cognitive Tutor Algebra dataset [28]. This dataset contains logs of student interactions with an automated algebra tutor. We collected all equations from the logs, and replaced their numerical constants by placeholders. This gave us 290 syntactic equation templates, such as

$$(\Box x + \Diamond) = x$$

and

$$(\Box - (-\Diamond)) = (((\bigstar/x) + (-\maltese)) - (-\blacklozenge)) \ .$$

To generate a problem, we first sample one of the templates, and then replace each constant independently by an integer between -10 and 10 inclusive, uniformly.

Table 4 lists all axioms in the domain, with examples of applying each.

The following are two examples of step-by-step solutions generated by ConPoLe for sampled problems, with the axioms used to derive each step. Numbers in square brackets represent fractions, not divisions (e.g. `[4/5]` means $\frac{4}{5}$).

```
(-7) = (3 - ((-7) / x)) =>
((-7) - 3) = ((3 - ((-7) / x)) - 3) | sub 3 =>
((-7) - 3) = ((3 - 3) - ((-7) / x)) | sub_comm 4, ((3 - ((-7) / x)) - 3) =>
((-7) - 3) = (0 - ((-7) / x)) | eval 5, (3 - 3) =>
(-10) = (0 - ((-7) / x)) | eval 1, ((-7) - 3) =>
-10x = ((0 - ((-7) / x)) * x) | mul x =>
(-10x / (-10)) = (((0 - ((-7) / x)) * x) / (-10)) | div (-10) =>
((x * (-10)) / (-10)) = (((0 - ((-7) / x)) * x) / (-10)) | comm 2, -10x =>
(x * ((-10) / (-10))) = (((0 - ((-7) / x)) * x) / (-10)) | assoc 1, ((x * (-10)) / (-10)) =>
(x * 1) = (((0 - ((-7) / x)) * x) / (-10)) | eval 3, ((-10) / (-10)) =>
x = (((0 - ((-7) / x)) * x) / (-10)) | mul1 1, (x * 1) =>
x = ((0x - (((-7) / x) * x)) / (-10)) | dist 3, ((0 - ((-7) / x)) * x) =>
x = ((0x - (x * ((-7) / x))) / (-10)) | comm 7, (((-7) / x) * x) =>
x = ((0x - ((x * (-7)) / x)) / (-10)) | assoc 7, (x * ((-7) / x)) =>
x = ((0x - (-7x / x)) / (-10)) | comm 8, (x * (-7)) =>
x = ((0 - (-7x / x)) / (-10)) | mul0 4, 0x =>
x = ((0 - ((-7) * (x / x))) / (-10)) | assoc 5, (-7x / x) =>
x = ((0 - ((-7) * 1)) / (-10)) | div_self 7, (x / x) =>
x = ((0 - (-7)) / (-10)) | eval 5, ((-7) * 1) =>
x = (7 / (-10)) | eval 3, (0 - (-7)) =>
x = ([-7/10]) | eval 2, (7 / (-10))
```

Table 4: Axioms of the `equations` domain.

| Mnemonic | Description | Example |
|---|---|---|
| refl | Reflexivity: if $a = b$, then $b = a$. | $1 + 2 = x \rightarrow x = 1 + 2$ |
| comm | Commutativity: $+$ and $\times$ commute. | $(2x)/2 = 4 \rightarrow (x \times 2)/2 = 4$ |
| assoc | Associativity: $+$ (resp. $\times$) associates over $+$ and $-$ (resp. $\times$ and $/$). | $((x + 1) - 1) = 9 \rightarrow (x + (1 - 1)) = 9$ |
| dist | Distributivity: $\times$ and $/$ distribute over $+$ and $-$. | $2 \times (x + 1) = 5 \rightarrow (2x + (2 \times 1)) = 5$ |
| sub_comm | Consecutive subtractions can have their order swapped. | $((2x - 1) - x) = 1 \rightarrow ((2x - x) - 1) = 1$ |
| eval | Operations with constants can be replaced by their result. | $x = (9/3) \rightarrow x = 3$ |
| add0 | Adding $0$ is an identity operation. | $(x + 0) = 9 \rightarrow x = 9$ |
| sub0 | Subtracting $0$ is an identity operation. | $(x - 0) = 9 \rightarrow x = 9$ |
| mul1 | Multiplication by $1$ is an identity operation. | $1x = 9 \rightarrow x = 9$ |
| div1 | Division by $1$ is an identity operation. | $(x/1) = 9 \rightarrow x = 9$ |
| div_self | Dividing a non-zero term by itself results in $1$ | $x = (5x/5x) \rightarrow x = 1$ |
| sub_self | Any term minus itself is $0$. | $x = ((x + 1) - (x + 1)) \rightarrow x = 0$ |
| subsub | Subtracting $-e$ is equivalent to adding $e$. | $(x - (-9)) = 10 \rightarrow (x + 9) = 10$ |
| mul0 | Multiplying by $0$ results in $0$. | $x = (1 + 0 \times 2x) \rightarrow x = (1 + 0)$ |
| zero_div | $0$ divided by a non-zero term results in $0$. | $x = (0/(x + 1)) \rightarrow x = 0$ |
| add | Any subterm can be added to both sides of the equation. | $(x - 1) = 0 \rightarrow ((x - 1) + 1) = (0 + 1)$ |
| sub | Any subterm can be subtracted from both sides of the equation. | $(x + 1) = 0 \rightarrow ((x + 1) - 1) = (0 - 1)$ |
| mul | Any subterm can be multiplied to both sides of the equation. | $(x/2) = 6 \rightarrow ((x/2) \times 2) = (6 \times 2)$ |
| div | Any subterm can be used to divide both sides of the equation. | $2x = 6 \rightarrow ((2x)/2) = (6/2)$ |

```
(2 + 8x) = (-2x + 10) =>
((2 + 8x) - -2x) = ((-2x + 10) - -2x) | sub -2x =>
((2 + 8x) - -2x) = ((10 + -2x) - -2x) | comm 11, (-2x + 10) =>
((2 + 8x) - -2x) = (10 + (-2x - -2x)) | assoc 10, ((10 + -2x) - -2x) =>
((2 + 8x) - -2x) = (10 + 0) | sub_self 12, (-2x - -2x) =>
(2 + (8x - -2x)) = (10 + 0) | assoc 1, ((2 + 8x) - -2x) =>
(2 + ((8 - (-2)) * x)) = (10 + 0) | dist 3, (8x - -2x) =>
(2 + 10x) = (10 + 0) | eval 4, (8 - (-2)) =>
(10x + 2) = (10 + 0) | comm 1, (2 + 10x) =>
((10x + 2) - 2) = ((10 + 0) - 2) | sub 2 =>
(10x + (2 - 2)) = ((10 + 0) - 2) | assoc 1, ((10x + 2) - 2) =>
(10x + 0) = ((10 + 0) - 2) | eval 5, (2 - 2) =>
10x = ((10 + 0) - 2) | add0 1, (10x + 0) =>
(10x / 10) = (((10 + 0) - 2) / 10) | div 10 =>
((x * 10) / 10) = (((10 + 0) - 2) / 10) | comm 2, 10x =>
(x * (10 / 10)) = (((10 + 0) - 2) / 10) | assoc 1, ((x * 10) / 10) =>
(x * 1) = (((10 + 0) - 2) / 10) | eval 3, (10 / 10) =>
x = (((10 + 0) - 2) / 10) | mul1 1, (x * 1) =>
x = ((10 - 2) / 10) | eval 4, (10 + 0) =>
x = (8 / 10) | eval 3, (10 - 2) =>
```

Table 5: Axioms of the `fractions` domain.

| Mnemonic | Description | Example |
|---|---|---|
| factorize | Factorize a composite integer into a prime factor times a divisor. | $\frac{20}{5} \rightarrow \frac{5 \times 4}{5}$ |
| cancel | Eliminate a common factor between both the numerator and the denominator. Only applies when the factor is explicitly written in both expressions. | $\frac{2 \times 5}{5 \times 10} \rightarrow \frac{2}{10}$ |
| eval | Evaluate an operation with numbers. | $\frac{2 \times 5}{10} \rightarrow \frac{10}{10}$ |
| scale | Multiply both the numerator and denominator of a fraction by a prime $p \in \{2, 3, 5, 7\}$. | $\frac{1}{2} + \frac{1}{6} \rightarrow \frac{1 \times 3}{2 \times 3} + \frac{1}{6}$ |
| simpl1 | Replace a fraction with denominator 1 by its numerator. | $\frac{10+5}{1} \rightarrow 10 + 5$ |
| mfrac | Rewrite a number as a fraction with denominator 1. | $5 + \frac{2}{3} \rightarrow \frac{5}{1} + \frac{2}{3}$ |
| mul | Multiply two fractions. | $\frac{3}{4} \times \frac{2}{3} \rightarrow \frac{3 \times 2}{4 \times 3}$ |
| combine | Add or subtract two fractions that have syntactically equal denominators. | $\frac{3}{4+1} + \frac{9 \times 2}{4+1} \rightarrow \frac{3+(9 \times 2)}{4+1}$ |

```
x = [4/5] | eval 2, (8 / 10)
```

## A.2 `fractions`

The `fractions` environment exercises the ability to reason about integer factorizations, especially common divisors and common multiples, from primitive axioms. A state in this environment is one of:

**Number:** An integer $n$,

**Number operations:** Either addition, subtraction or multiplication of two terms, both of which can be either numbers or number operations,

**Fraction:** A single fraction, where numerator and denominator are either numbers or number operations,

**Fraction operation:** An operation $(+, -$ or $\times)$ between two fractions, two numbers, or a fraction and a number.

A state is solved if it is either a number or a fraction where both numerator and denominators are numbers that are coprime (i.e. their greatest common divisor has to be 1). Note that a fraction operation can only involve two fractions, not other (recursively defined) fraction operations. This is to keep this domain testing an orthogonal skill compared to `equations`: nested operations would require more elaborate algebraic manipulations, but this environment focuses on the Common Core topic of fraction manipulation.

The following are three random problems solved by ConPoLe demonstrating all axioms:

```
[1]/[105] + [1]/[42] =>
[1]/[105] + [(5 * 1)]/[(5 * 42)] | scale 4, 5 =>
[1]/[105] + [(5 * 1)]/[210] | eval 8, 5 * 42 =>
[(2 * 1)]/[(2 * 105)] + [(5 * 1)]/[210] | scale 1, 2 =>
[(2 * 1)]/[210] + [(5 * 1)]/[210] | eval 5, 2 * 105 =>
[((2 * 1) + (5 * 1))]/[210] | combine 0 =>
```

Table 6: Axioms of the `ternary-addition` domain.

| Mnemonic | Description | Example |
|----------|-------------|---------|
| swap | Swap any two adjacent digits | $b3\ b5\ c3 \to b3\ c3\ b5$ |
| comb | Combine (add) two adjacent digits that multiply the same power $p$, replacing them by two other digits: the result (which has power $p$) and the carry (with power $p + 1$). | $b3\ c3\ b5 \to a3\ b4\ b5$ |
| del | Erase a digit 0 ($a$). | $a3\ b4\ b5 \to b4\ b5$ |

```
[(2 + (5 * 1))]/[210] | eval 2, 2 * 1 =>
[(2 + 5)]/[210] | eval 3, 5 * 1 =>
[7]/[210] | eval 1, 2 + 5 =>
[7]/[(7 * 30)] | factorize 2, 210, 7*30 =>
[1]/[30] | cancel 0, 7

[18]/[5] - 1 =>
[18]/[5] - [1]/[1] | mfrac 4, 1 =>
[18]/[5] - [(5 * 1)]/[(5 * 1)] | scale 4, 5 =>
[18]/[5] - [5]/[5] | cancel 4, 1 =>
[(18 - 5)]/[5] | combine 0 =>
[13]/[5] | eval 1, 18 - 5

5 * 3 =>
[5]/[1] * 3 | mfrac 1, 5 =>
[5]/[1] * [3]/[1] | mfrac 4, 3 =>
[(5 * 3)]/[(1 * 1)] | mul 0 =>
[(5 * 3)]/[1] | eval 4, 1 * 1 =>
[15]/[1] | eval 1, 5 * 3 =>
15 | simpl1 0
```

We used a custom generator for fraction problems. First, with 25% chance, we choose to generate a single-term problem; otherwise, we will generate a *fraction operation* (two terms with an operator drawn uniformly from $\{+, -, \times\}$). We then generate the subterms independently as follows. With 50% chance, we generate a number. A number is generated by first picking the number of prime factors (between 0 and 4), then drawing each factor independently from the set $\{2, 3, 5, 7\}$ and multiplying them. A fraction is generated by generating two numbers with the same described procedure: the first becomes the numerator, and the second becomes the denominator.

### A.3 `ternary-addition`

The `ternary-addition` domain exercises step-by-step arithmetic, in an analogous fashion to some example-tracing arithmetic tutors [30], where operations can be performed out of the traditional order as long as they are correct deductions. Each state is a sequence of digits multiplying powers of 3, that are being added together. Two digits can be combined (added together) when they are adjacent and multiply the same power (e.g. $2 \times 3^3$ and $1 \times 3^3$ can be combined together, but $2 \times 3^3$ and $1 \times 3^5$ cannot). Three operations are available: (a) combining two adjacent digits that multiply the same power – generating two other digits, (b) swapping any pair of adjacent digits, and (c) deleting a digit 0 from anywhere. A state is solved when the final number can be readily read from the state: all digits must multiply different powers, they must be sorted by power, and there should be no zero digits. For example, $2 \times 3^3 + 1 \times 3^5$ is simplified. On the other hand, $2 \times 3^3 + 1 \times 3^5 + 1 \times 3^3$ is not: the digits multiplying $3^3$ can be brought together and further combined.

To represent digits and powers as strings, we use the letters $a, b, c$ to represent digits $0, 1, 2$ respectively, and decimal digits $0 - 9$ to represent powers. There is an implicit addition operation between all digits in the state. For example, $c3\ b5\ b3$ represents $2 \times 3^3 + 1 \times 3^5 + 1 \times 3^3$. Table 6 lists the three axioms described above, with examples.

The following are two of ConPoLe's solutions for random problems, both utilizing all 3 axioms.

Table 7: Axioms of the `sorting` domain.

| Mnemonic | Description | Example |
|---|---|---|
| swap | Swap two adjacent elements. | [=\|==\|====\|===] → [=\|==\|===\|====] |
| reverse | Reverse the entire list. | [===\|==\|=] → [=\|==\|===] |

```
#(c3 c3 b5 b5 b5 a1 a0 c0) =>
#(c3 c3 b5 c5 a6 a1 a0 c0) | comb 3, b5 b5 =>
#(c3 c3 a5 b6 a6 a1 a0 c0) | comb 2, b5 c5 =>
#(c3 c3 a5 b6 a6 a1 c0) | del 6, a0 =>
#(c3 c3 a5 b6 a6 c0) | del 5, a1 =>
#(c3 c3 a5 b6 c0) | del 4, a6 =>
#(c3 c3 a5 c0 b6) | swap 3, b6 c0 =>
#(c3 c3 c0 b6) | del 2, a5 =>
#(c3 c0 c3 b6) | swap 1, c3 c0 =>
#(c0 c3 c3 b6) | swap 0, c3 c0 =>
#(c0 b3 b4 b6) | comb 1, c3 c3

#(a1 b5 c1 b3 c3 b5 a2 c1 c1 c1 b0 b3 a5 b5) =>
#(a1 b5 c1 b3 c3 b5 a2 c1 b1 b2 b0 b3 a5 b5) | comb 8, c1 c1 =>
#(a1 b5 c1 b3 c3 b5 a2 a1 b2 b2 b0 b3 a5 b5) | comb 7, c1 b1 =>
#(b5 c1 b3 c3 b5 a2 a1 b2 b2 b0 b3 a5 b5) | del 0, a1 =>
#(b5 c1 b3 c3 b5 a2 b2 b2 b0 b3 a5 b5) | del 6, a1 =>
#(b5 c1 a3 b4 b5 a2 b2 b2 b0 b3 a5 b5) | comb 2, b3 c3 =>
#(b5 c1 b4 b5 a2 b2 b2 b0 b3 a5 b5) | del 2, a3 =>
#(c1 b5 b4 b5 a2 b2 b2 b0 b3 a5 b5) | swap 0, b5 c1 =>
#(c1 b5 b4 b5 b2 b2 b0 b3 a5 b5) | del 4, a2 =>
#(c1 b5 b4 b5 c2 a3 b0 b3 a5 b5) | comb 4, b2 b2 =>
#(c1 b4 b5 b5 c2 a3 b0 b3 a5 b5) | swap 1, b5 b4 =>
#(c1 b4 c5 a6 c2 a3 b0 b3 a5 b5) | comb 2, b5 b5 =>
#(c1 b4 c5 c2 a3 b0 b3 a5 b5) | del 3, a6 =>
#(c1 b4 c5 c2 b0 b3 a5 b5) | del 4, a3 =>
#(c1 b4 c5 b0 c2 b3 a5 b5) | swap 3, c2 b0 =>
#(c1 b4 b0 c5 c2 b3 a5 b5) | swap 2, c5 b0 =>
#(c1 b0 b4 c5 c2 b3 a5 b5) | swap 1, b4 b0 =>
#(b0 c1 b4 c5 c2 b3 a5 b5) | swap 0, c1 b0 =>
#(b0 c1 b4 c5 c2 b3 b5) | del 6, a5 =>
#(b0 c1 b4 c2 c5 b3 b5) | swap 3, c5 c2 =>
#(b0 c1 b4 c2 b3 c5 b5) | swap 4, c5 b3 =>
#(b0 c1 b4 c2 b3 a5 b6) | comb 5, c5 b5 =>
#(b0 c1 b4 c2 b3 b6) | del 5, a5 =>
#(b0 c1 c2 b4 b3 b6) | swap 2, b4 c2 =>
#(b0 c1 c2 b3 b4 b6) | swap 3, b4 b3
```

To generate a problem, we first pick the number of digits in the sequence uniformly from 1 to 15. Then, we choose each element independently, by choosing a digit from $\{0, 1, 2\}$ and a power from $\{0, 1, 2, 3, 4, 5, 6\}$, all independently and uniformly.

## A.4 `sorting`

The `sorting` environment tests the ability to measure and compare object lengths, inspired by the "Measurements and Data" section from Common Core. States in this domain are a permutation of the integers from 1 to $L$, where $L$ is the length of the list. When represented as a string, each number $n_i$ is written as a repetition of the = character, $n_i$ times; | is used as a separator between numbers. The goal is to sort the list by the length of each of the substrings. Table 7 lists the only two axioms in this domain: swapping adjacent elements and reversing the list. Below, we show two solutions generated by ConPoLe. The first is done with swaps only. In the second problem, the reversed list has less inversions than the given one: ConPoLe learns to first reverse the list, and then sort the result using swaps.

```
[====|==|=|===|=====|======] =>
```

```
[====|=|==|===|=====|======] | swap 1 =>
[=|====|==|===|=====|======] | swap 0 =>
[=|==|====|===|=====|======] | swap 1 =>
[=|==|===|====|=====|======] | swap 2

[========|======|===|=|==|====|=======|=====]
[=====|=======|====|==|=|===|======|=======] | reverse =>
[=====|====|======|==|=|===|======|=======] | swap 1 =>
[=====|====|==|======|=|===|======|=======] | swap 2 =>
[=====|====|==|=|======|===|======|=======] | swap 3 =>
[=====|====|==|=|===|======|======|=======] | swap 4 =>
[=====|==|====|=|===|======|======|=======] | swap 1 =>
[=====|==|====|=|===|======|======|=======] | swap 5 =>
[=====|==|=|====|===|======|======|=======] | swap 2 =>
[=====|==|=|===|====|======|======|=======] | swap 3 =>
[=====|=|==|===|====|======|======|=======] | swap 1 =>
[=|=====|==|===|====|======|======|=======] | swap 0 =>
[=|==|=====|===|====|======|======|=======] | swap 1 =>
[=|==|===|=====|====|======|======|=======] | swap 2 =>
[=|==|===|====|=====|======|======|=======] | swap 3
```

For generating problems, we first choose a length $L$ uniformly from 2 to 11, and then shuffle the list of integers from 1 to $L$. Lists of 11 elements have at most 55 inversions. Therefore, because of the `reverse` operation, all of them can be sorted with at most 27 adjacent swaps (plus one use of `reverse`, potentially).

## B  Training and architecture details

**Training/test split**. The generators described in Appendix A use pseudo-random number generators, and thus are deterministic if the random seed is fixed. We use this fact to generate distinct training and test environments. For training, agents start with a random seed given by a OS-provided randomness source. Every time an agent samples a new problem, a new seed is chosen from $10^6$ to $10^7$ (providing around $10^7$ potential training problems). For testing, we always use the seeds from 0 to 199, providing 200 training problems.

**Architecture details**. All models use character-level bidirectional LSTM encoders. We first use 64-dimensional character embeddings. Then, we use two stacked bi-LSTM layers, with a hidden dimension of 256. Finally, we take the last hidden state of each direction from the last layer, concatenate their vectors to obtain a 512-dimensional embedding of the state, and transform this embedding with a 2-layer MLP that preserves dimension (and do the same separately for the action, in DRRN), and use that final output according to each model's architecture. ConPoLe learns a $512x512$ matrix $W_\theta$ that performs the bilinear transform; CVI learns a linear layer, and DRRN embeds state and action and outputs their dot product.

**Hyper-parameters**. We first picked the learning rate from $10^{-i}$ and $5 \times 10^{-i}$ for $i$ ranging from 1 to 6; in shorter experiments of 100k environment steps in `equations` and `fractions`, the value of $5 \times 10^{-6}$ had the highest success rate for CVI and ConPoLe (though the difference to $10^{-5}$ and $5 \times 10^{-5}$ was insignificant); for DRRN, $10^{-4}$ performed best on average. We thus used these values in all experiments. Next, we picked the frequency of updates and batch sizes. For ConPoLe and CVI, we observed that more frequent updates were consistently better; for performance, we chose to optimize every 10 solved problems, taking 256 gradient steps on randomly sampled contrastive examples from the replay buffer. For DRRN, since each training example requires computing a $\max$ operation for the $Q$ update, we chose a smaller batch size to keep training runs in a single domain under 3 days. We therefore picked a batch size of 64, and performed training updates every 16 problems.