# OpenReview forum: "Contrastive Reinforcement Learning of Symbolic Reasoning Domains"
_NeurIPS.cc/2021/Conference — NeurIPS 2021 Poster_

### Official Review · Reviewer_cgrc · 2021-07-16

**Rating:** 5
**Confidence:** 4

**Summary:**

In this paper, the authors address the problem of learning policies on symbolic domains - like mathematical equations. They propose ConPoLe - learning algorithm that relies on optimizing InfoNCE, a well-known contrastive loss. They benchmark ConPoLe on four different environments and show that it generalizes well. A key contribution of the paper is the introduction of these environments themselves.

**Limitations And Societal Impact:**

Yes.

**Main Review:**

The primary premise of this paper is to learn effective policies on symbolic domains. To that end, the authors first create four different environments drawn from the Mathematics curriculum of the US Common Core State Standards. Their solution approach is motivated by the observation that learning value estimates is not difficult when rewards are sparse. They focus specifically on symbolic domains that are characterized by a single reward state - e.g., Rubik's cube. Their approach relies on minimizing the InfoNCE loss and avoids value estimation completely. This differentiates their approach from other works in this area - notably DeepCube.

Strengths:
1. The four new environments introduced in this paper are valuable to the RL community. Since they are drawn from a typical mathematics curriculum, solutions in this space might be expected to improve educational tools

2. Formulating this problem in terms of a contrastive learning problem is somewhat novel - and effectively sidesteps the need to calculate noisy value function estimates when the rewards are sparse as in these environments.

Weaknesses:
1. The paper relies quite a bit on datasets and reinforcement learning baselines that the authors develop. As such, it is difficult to ascertain the effectiveness of their approach over modern learning methods. For example, for the CommonCore environments, 3 out of the 4 methods evaluated are variations of ConPoLe - with DRRL, a relatively old method, being the only baseline.

A comparison between ConPoLe and Autodidactic Iteration - the approach used in DeepCube - on all the environments would have made the paper much stronger.

2. The more interesting learning problem - from the point of view of scale - is the comparison with DeepCube. The paper doesn't provide any insight into whether ConPoLe is actually competitive. The only datapoint provided is that ConPoLe takes a significantly higher number of average-number-of-moves compared to DeepCube while also visiting less than half the number of nodes. This single datapoint is hardly adequate to lend credibility to their claim that "this result validates the generality and promise of ConPoLe for solving challenging symbolic domains".

Post-rebuttal comments: Thanks to the authors for reporting on the additional experiments. Based on the response, I have increased my score to 5. The comparison between ConPole and ADI makes this paper stronger. However, I still have concerns on the general applicability of ConPole to larger scale problems compared to the datasets the authors introduce. I think this is a good paper in the making if the authors are able to compare ADI and ConPole on a more complex problem like Rubik's cube.

**Time Spent Reviewing:**

1

---

> ### Author Response · Authors · 2021-08-10
> **Response and new comparison to Autodidatic Iteration**
>
> We thank the reviewer for the time and encouragement regarding the Common Core domains. We attempt to address the main concerns regarding the comparisons below.
>
> > with DRRL, a relatively old method, being the only baseline. A comparison between ConPoLe and Autodidactic Iteration - the approach used in DeepCube - on all the environments would have made the paper much stronger.
>
> We appreciate the suggestion of running Autodidatic Iteration (ADI) on all environments. We performed this comparison. Please refer to the general comment for a detailed discussion on the results, in brief ADI is a stronger baseline than DRRL but does not approach the performance of ConPoLe. We will add these results to the paper.
>
> > The more interesting learning problem - from the point of view of scale - is the comparison with DeepCube. The paper doesn't provide any insight into whether ConPoLe is actually competitive.
>
> Our observation was that ConPoLe was able to learn a working heuristic that enables A* to solve arbitrary instances of the Rubik's Cube. By itself, we believe this is a non-trivial result: two other specialized expert-written solvers time out on instances with 1000 scrambles (Korf and Rokiki), as observed by the DeepCube authors. Heuristic solvers like Kociemba always work, but developing one requires deep knowledge of the cube. Even if we do not claim state-of-the-art results in Rubik's Cube specifically, our key observation was that we were able to produce a general working solver with minimal effort with the general ConPoLe loss, without directly optimizing for search.

---

> ### Author Response · Authors · 2021-08-16
> **Updated opinion?**
>
> We would be very grateful if reviewer cgrc could provide us with an updated opinion (and updated rating if applicable) given the additional results described in the general response? In particular, they said "A comparison between ConPoLe and Autodidactic Iteration - the approach used in DeepCube - on all the environments would have made the paper much stronger."; we have reported this comparison above and hope that it does indeed make the paper much stronger!

---

> > ### Comment · Reviewer_cgrc · 2021-08-16
> > **Comparison with ADI on Common Core tasks look good**
> >
> > Thank you for providing the updates on running ADI on the common core datasets. This definitely alleviates my concern #1 on baselines quite a bit and I will be increasing my score based on that. I will update my review and scores based on discussions with other reviewers.

---

### Official Review · Reviewer_MxYv · 2021-07-17

**Rating:** 7
**Confidence:** 3

**Summary:**

This paper proposes a new reinforcement learning method for symbolic domains with sparse binary rewards (e.g., finding step-by-step solutions to math problems). The method assumes the availability of many problem instances within the domain, of varying levels of difficulty; the agent learns by solving the easier problems first, and training to distinguish positive and negative examples of state transitions on the way to a solution, using a version of the InfoNCE loss from the contrastive learning literature. ConPoLe operates directly on text representations of states within a domain, so adapting it to new domains does not require domain-specific hand-engineered features. The paper presents four novel domains corresponding to different kinds of math problem, inspired by the Common Core curriculum, and shows that ConPoLe outperforms existing methods by a wide margin. In addition, a small-scale experiment suggests that the embeddings that ConPoLe learns cluster according to existing pedagogical classifications of math problems on Kahn Academy. ConPoLe is also tested in an established Rubik's cube domain, comparing favorably to the DeepCubeA baseline in terms of data efficiency and search time (if not final solution length).

**Limitations And Societal Impact:**

Yes, I think limitations and societal impact are adequately addressed.

**Main Review:**

ConPoLe is an appealingly simple idea. It exploits the unique features of the sorts of symbolic domains considered here (unlimited problem examples, of varying levels of difficulty), and avoids relying on meaningful intermediate rewards or hand-engineered features, which would be cumbersome to develop. The experiments are compelling and even if ConPoLe does not end up scaling to domains like program synthesis and automated theorem proving, I buy that there are already important application areas (like interactive tutoring systems) in which ConPoLe could be really useful.

I have two suggestions for improving the clarity of the paper, and two questions that would be great to see addressed in the paper (or perhaps as part of follow-on work). First, the suggestions:

1. Although "beam search" is a standard algorithm, there are different ways of instantiating it; I had trouble figuring out how exactly yours worked. Suppose at step $i$ there are $N$ trajectories $(s^j_1, \dots, s^j_i)$ (for $j = 1, \dots, N$) in the beam. Presumably, the beam is extended by considering all valid extensions-by-one-state of the current trajectories—for each trajectory in the beam, for each action, we can run the action to determine an extended trajectory. But how are these extended trajectories scored, to then cull all but the top $N$? I thought the score might be the probability, under the current policy, of the entire sequence (where the policy is to select a next state s' proportionally to f(s^j_i, s'), across the *valid* next state). But I wasn't sure, and spelling this out would be a great help. (Hopefully I haven't gravely misunderstood!)

It was also unclear to me exactly how the negative examples were chosen. When beam search concludes, we have $N$ trajectories of length, say, $k$ in the beam, with one of them ending in success. For each state $s_i$ along the successful trajectory, which negative examples are added? The pseudocode makes it seem like only the $(i+1)$th state of each trajectory (excluding the successful one) is included as a negative example. But why? These $(i+1)$th states are parts of trajectories that don't necessarily include the successful trajectory's $s_i$; what makes them more appropriate as negative examples than any of the other states in the replay buffer?

2. I wasn't sure what to make of the "mutual information" intuition in L189-196. The claim seems to be that the loss is a bound on the mutual information between successive states in successful trajectories. Does this argument depend at all on the distribution from which the negative examples are generated? A cursory skim of [22]'s notation makes it look as though they should be generated from a marginal distribution, e.g. the marginal distribution over states found along successful trajectories for problems sampled from your "problem generator." Does generating negatives via beam search actually sample from this distribution? L195 seems to suggest that your Section 5 observation that using more negatives improves learning is connected to the tightness of the InfoNCE bound on mutual information, but I wasn't sure why. What is the connection between a tighter bound on mutual information, and learning quality? My current impression is that the paper would be fine without this paragraph, but if it is included, it would be great if you could unpack the intuition a bit more.

My two general questions:

1. It's very nice that ConPoLe doesn't require hand-engineered feature representations. But I am curious about the extent to which it relies on correlational information present in the textual representations of states. Do you think the particular state representations you use meaningfully affect performance? Do you think pretraining / sharing the embeddings across different domains could help?

2. How important is it to get the data-generating distribution right during training? I can imagine a few ways a task might be 'out-of-distribution' -- e.g., if it requires a longer sequence of steps than the tasks seen during training, or is simply a longer string (more arithmetic operations in the equation, e.g.). When would ConPoLe succeed and fail at generalizing? Have you tested its ability to generalize in this way (or other ways)?

**Time Spent Reviewing:**

5 hours

---

> ### Author Response · Authors · 2021-08-10
> **Responses to questions and comments**
>
> We thank the reviewer for the positive feedback and encouraging comments about the problem setup and potential applications of ConPoLe! We hope the responses below help clarify the confusing points.
>
> > Although "beam search" is a standard algorithm, there are different ways of instantiating it; I had trouble figuring out how exactly yours worked.
>
> We believe your description matches exactly what we do: the partial trajectories are scored by their (log) probabilities since the starting state, and we then keep the top-k. This certainly deserves to be written out more clearly in the text.
>
> > It was also unclear to me exactly how the negative examples were chosen.
>
> When beam search is successful, for each intermediate step i we know one next step that leads to the solution. At the same time, during step i, many expanded next states were produced, either by the i-th state in the solution or by other states in the i-th beam. We use all of those as negatives corresponding to the positive pair of states i and i+1 in the solution. Thus, given the i-th state in the solution, this contrastive example consists of correctly choosing the solution state i+1 considering that state between all other states expanded during step i of beam search.
>
> You do make a good point that we could potentially try a comparison with using random states from the replay buffer as negatives. Our intuition is that the negative examples coming from the same problem will likely be harder to discriminate than random examples with obviously unreachable states. Given theoretical results on hard negative example mining for contrastive learning, if that is true, it could lead to representations that capture more information. However, there might still be benefit in adding random negatives on top, especially in harder domains or for states where there aren't enough expanded states.
>
> > I wasn't sure what to make of the "mutual information" intuition in L189-196.
>
> The main idea of making the connection to mutual information was to provide a meaning for the discriminative loss in our case. Since "mutual information" can simply be understood as "uncertainty reduction", the more bits of mutual information we capture (i.e. the lower the negative InfoNCE loss), the less uncertain (as in entropy) the agent's predictions will be. The bound itself is not strictly necessary for empirical performance, but it serves as an interpretation of the optimization in this view of RL as contrastive learning. For the original NCE bound to hold, you're right that the negatives need to be sampled from the marginal state distribution. However, there is theoretical justification for using "hard negatives", i.e. negatives that lie closer to the current state in embedding space (indeed, [30] shows one strategy that still preserves the bound). We should indeed clarify this imprecision in the paper if we end up keeping the paragraph you cited.
>
> > It's very nice that ConPoLe doesn't require hand-engineered feature representations. But I am curious about the extent to which it relies on correlational information present in the textual representations of states. Do you think the particular state representations you use meaningfully affect performance?
>
> We believe the particular state representations are likely to not be extremely important, as long as they contain the relevant information for making predictions. This intuition comes most strongly from the Rubik's Cube domains, in which our string representation of the cube looked particularly uninformative to us, but still seemed to be enough for ConPoLe with a simple encoder. We did, at different points, have two separate versions of the equations environment with slightly different string representations (especially regarding when to use parentheses and implicit multiplications, like 2*x instead of 2x), but we didn't notice any difference in performance when learning.
>
> > Do you think pretraining / sharing the embeddings across different domains could help?
>
> This idea is very much in line with follow-up experiments we want to run. One advantage of string-based environments is indeed, as you suggested, that we could potentially transfer learning across domains. We had some initial encouraging results in a multi-task setting, where a single agent had to learn multiple domains at once, and for some subsets of domains we did observe a speed-up in learning, suggesting transfer was happening. However, we decided to investigate and understand these results in more depth before presenting them. But we're very excited that these new domains might allow the community to further investigate this direction!
>
> > How important is it to get the data-generating distribution right during training?
>
> Unlike the string representation, our intuition is that the generator distribution is indeed quite important. As we touch on in Section 3, there needs to be an implicit curriculum, as we seek to bootstrap from easy problems, that should be solvable by an untrained agent, to harder ones. Intuitively, there shouldn't be too abrupt difficulty gaps in the distribution, although we observed ConPoLe to be more robust to this than Autodidatic Iteration (please see our general comment about this new comparison).
>
> > When would ConPoLe succeed and fail at generalizing? Have you tested its ability to generalize in this way (or other ways)?
>
> This also touches on one point we wish to explore further in future work. We believe these environments allow for quite interesting systematic generalization tests, as the one you suggested (training in easier problems; evaluating in harder ones). We have not yet performed such experiments. We do have one anecdotal data point, though. ConPoLe was able to solve one of the Khan Academy equations even if it had a simple feature that was missing from the training set: the variable was being divided by an expression involving constants. In the Cognitive Tutor Algebra dataset, there are many instances of the variable being multiplied by constant expressions, and also of constants being divided by the variable, but surprisingly not the opposite. However, we believe ConPoLe succeeded simply because it had seen variables divided by constants as intermediate states to many problems. This suggests that intermediate states can make up for some gaps in the data-generating distribution.
>
> We'd like to thank the reviewer again for the constructive feedback and all the thoughtful comments and questions!

---

> > ### Comment · Reviewer_MxYv · 2021-08-29
> > **Thanks for these responses**
> >
> > Thank you for these thorough responses to my questions.
> >
> > A couple brief notes in reply:
> >
> > > The main idea of making the connection to mutual information was to provide a meaning for the discriminative loss in our case. Since "mutual information" can simply be understood as "uncertainty reduction", the more bits of mutual information we capture (i.e. the lower the negative InfoNCE loss), the less uncertain (as in entropy) the agent's predictions will be.
> >
> > Thank you for the clarification. Perhaps the missing link that could be explained more fully in a revision is how the bound on the mutual information (the true value of which is fixed by the problem domain) is related to the agent's policy. (E.g., does the looseness of the bound quantify the agent's 'excess' uncertainty, which optimization reduces?)
> >
> > > Our intuition is that the negative examples coming from the same problem will likely be harder to discriminate than random examples with obviously unreachable states.
> >
> > To clarify, I didn't mean to add negative examples from the entire reply buffer. It seems there are a few different levels of selectiveness you could use:
> >
> > 1. For a state s_i in a successful trajectory, use _only_ negative examples expanded starting from s_i
> > 2. For a state s_i in a successful trajectory, use negative examples expanded from any s'_i -- i.e., expanded starting from the step-i state of another trajectory in the beam [it sounds like this is what you do]
> > 3. For a state s_i in a successful trajectory, use negative examples generated at any point during the beam search *for this problem*
> > 4. For a state s_i in a successful trajectory, randomly sample negative examples from the entire domain
> >
> > I'm curious about the intuition for choosing (2) instead of (3), given that a priori I don't see a reason to think s'_i (for some other trajectory s') should be particular similar to s_i.

---

> > > ### Author Response · Authors · 2021-09-02
> > > **Thank you, and further clarifications**
> > >
> > > Thank you for the responses! We try to clarify these last points below.
> > >
> > > > Perhaps the missing link that could be explained more fully in a revision is how the bound on the mutual information (the true value of which is fixed by the problem domain) is related to the agent's policy. (E.g., does the looseness of the bound quantify the agent's 'excess' uncertainty, which optimization reduces?)
> > >
> > > This is indeed a valid interpretation -- in a perfectly tight bound, the only uncertainty remaining is due to the environment (in this case, since the environments are deterministic, the only uncertainty arises from there being multiple possible solutions). Optimization then drives the policy towards this direction. We will make this point more clear in a revision.
> > >
> > > > I'm curious about the intuition for choosing (2) instead of (3), given that a priori I don't see a reason to think s'_i (for some other trajectory s') should be particular similar to s_i.
> > >
> > > I see, thanks for clarifying. At the beginning, it is indeed the case that trajectories in the beam do not look similar at all. In the course of training, however, we did observe a higher similarity between states in the beam due to necessary steps that don't visibly change the state too much. In equations, for instance, the solutions can be usually split into two kinds of segments: (a) a few key actions that clearly advance the solution (usually evaluating a constant sub-expression or adding/subtracting/multiplying something on both sides), intertwined with (b) sequences of algebraic manipulations that eventually make the next key action possible (this is usually applying commutativity/associativity multiple times, moving terms and parentheses around). During (b), the state as a string doesn't change significantly when the operations are picked sensibly. The trained agent tends to put similar-looking operations in the beam (with confidence concentrating on the actually useful one), since our beam search doesn't enforce diversity. This was the original motivation for choosing (2) as negatives.
> > >
> > > That said, (3) should perform at least as well due to it being a superset of (2), and thus a harder discrimination problem. The only drawback might be the performance drop in encoding more negatives. However, it would indeed be interesting to analyze if (3) speeds up training as a function of environment steps, ignoring latency.

---

### Official Review · Reviewer_xzcf · 2021-07-17

**Rating:** 6
**Confidence:** 4

**Summary:**

This paper proposes a RL approach (ConPoLe) that minimizes the contrastive loss to tackle some elementary school level symbolic reasoning problems (CommonCore) like solving level simple math equations, fractions, and sorting, where the state is represented as a general string rather than as structured input (e.g. AST after parsing). The problem is difficult because the reward is sparse (only rewarded when successed). The main idea is to learn a parametric score function for next state proposals, through contrastive learning treating the one that led to the solution as positive and the rest ones as negative examples. The proposed method can achieve successful results on those four symbolic reasoning domains, outperforming all baselines.


**Limitations And Societal Impact:**

Limitation of the work is discussed. There is no concerns or issues, at least to my understanding, about negative social impact.

**Main Review:**

This paper studies an interesting and novel problem and application of RL. I think being able to solve complex reasoning problems (such as automatic proof) is in general a very interesting and important area of study. As such, introducing four symbolic reasoning domains seems to be a good novelty of this work and I think this can be a valuable environment for the community. Empirical results are also promising.

However, I have some concerns about the paper. I would appreciate authors to address these questions and improve writing of the paper in general.

First of all, I am not sure this approach can be seen as an instance of reinforcement learning; this looks more like conventional AI or state search algorithm with a potential/score function learned through contrastive representation learning. There is not much MDP formalism (like reward, policy, or value function) being involved -- the policy or planner for solving a task at hand (i.e. test setting) is simply a beam search with trying all possible actions.

In terms of method/technical approach, search of solution space is done via beam search.
One limitation therefore is that it cannot learn from unsolved problems via beam search. This might be okay given that we are able to find some solution with an uninformed but reasonable beam search or IDA like search algorithm, but the method won’t scale up and work well on difficult reasoning tasks (e.g. automatic proof) where random or exhaustive MCTS is not enough.

Arguably, the environment is too simple. Compared to automatic proof and solving logic problems, the problem appears to be solvable with a reasonable heuristic or efficient state space search algorithm (such as A* or IDA).

The authors claim the state observation is given as an unstructured string. However, it seems that during the next state generation through possible action(mnemonic) some semantic structure is taken into account. In order to apply those actions and tell the operand (e.g. sub “3”, sub_comm “4”), either the agent or the environment itself should be aware of the structure of the equation. So claiming the environment is essentially “unstructured string” might sound like a bit of overclaiming.


Writing/Clarity:

- Qualitative examples or demo of the environment/task needs to be given in the main paper without need to look at the supplementary.
- Section 4 seems written verbose. It would be a much more clear organization if section 4 had several subsections (or paragraphs) describing each component constituting the overall algorithm.
- Algorithm 1: the authors should avoid using weird italic fonts in LaTeX math environments -- roman fonts preferred.
- In Algorithm 1, $\theta_\pi$ is the only learnable parameter, but this is mentioned nowhere in the main text. In the score function formulation (L173-180), we have $\theta$ --- are they essentially the same thing? Also, it’s not clear how exactly $f_\theta$ is being incorporated into the policy. It is also not very clear what is the exact policy setup for exploitation (i.e. how to solve the given task during test after training is done); do we still use beam search?
- Figure 2 seems to be placed in a random place. It would be good to place them in the top area.


----------

POST-REBUTTAL UPDATE: I have updated my rating from 5 to 6. Please see my comments below in reply to the author feedback.

**Time Spent Reviewing:**

5 hours; additional ~10 hours for post-rebuttal discussions

---

> ### Author Response · Authors · 2021-08-10
> **New comparions and discussions**
>
> We thank the reviewer for the encouragement regarding the problem and the domains we propose. We address specific concerns and questions below:
>
> > First of all, I am not sure this approach can be seen as an instance of reinforcement learning; this looks more like conventional AI or state search algorithm with a potential/score function learned through contrastive representation learning. There is not much MDP formalism (like reward, policy, or value function) being involved -- the policy or planner for solving a task at hand (i.e. test setting) is simply a beam search with trying all possible actions.
>
> We mention in Section 5.2, but perhaps don't emphasize it enough, that during test time we don't perform search: we take the agent's top prediction only at each state, and perform a single roll-out per problem using this greedy policy. Reward is given if the problem is solved. In terms of the MDP structure and how we see, this RL formulation is similar to other works on reinforcement learning for theorem proving: a binary (or discrete) reward is given once proof goals are met, and agents have a variable-sized action pool in each state. However, in the Common Core domains we don't use beam search/MCTS at test time, and rather expect agents to solve problems with a single roll-out (as a human expert would have no trouble doing).
>
> > In terms of method/technical approach, search of solution space is done via beam search. One limitation therefore is that it cannot learn from unsolved problems via beam search. This might be okay given that we are able to find some solution with an uninformed but reasonable beam search or IDA like search algorithm, but the method won’t scale up and work well on difficult reasoning tasks (e.g. automatic proof) where random or exhaustive MCTS is not enough.
>
> We note that using some flavor of stochastic search is also what prior work on formal mathematical domains typically do (e.g. MCTS in leanCoP, beam search in ASTatic). In general, in line with the intuition you described, to learn in the interactive setting (i.e., without human solutions) there must be an implicit curriculum in the problems, with some easy problems solvable from an uninformed stochastic search, so that the agent can then bootstrap to harder problems.
>
> > Arguably, the environment is too simple. Compared to automatic proof and solving logic problems, the problem appears to be solvable with a reasonable heuristic or efficient state space search algorithm (such as A* or IDA).
>
> Please refer to our general comment above for a deeper comparison to A*/IDA, Google MathSteps (the library we cited which implements solving heuristics for equations), and Autodidatic Iteration (the DeepCube algorithm). In brief, we believe that the problems we pose are not easily solvable by these methods.
>
> > The authors claim the state observation is given as an unstructured string. However, it seems that during the next state generation through possible action(mnemonic) some semantic structure is taken into account. In order to apply those actions and tell the operand (e.g. sub “3”, sub_comm “4”), either the agent or the environment itself should be aware of the structure of the equation. So claiming the environment is essentially “unstructured string” might sound like a bit of overclaiming.
>
> Yes, this is a great point for us to clarify! The states are unstructured strings only from the perspective of the agent/model (which operates with the exact same string-based interface with all environments, including equations and the Rubik's Cube). From the environment's perspective, the strings do have a known structure. In fact, each environment has its own parser and representation, which it uses to produce candidate actions. Thus we are interested in more general agent models, while being willing to build more knowledge into the environment specification that the agent will learn from. We will make this distinction more clear in the writing.
>
> > In Algorithm 1, is the only learnable parameter, but this is mentioned nowhere in the main text. In the score function formulation (L173-180), we have --- are they essentially the same thing?
>
> Thanks for the catch, these parameters are indeed the same thing. We will fix this inconsistency in the text.
>
> > Also, it’s not clear how exactly is being incorporated into the policy. It is also not very clear what is the exact policy setup for exploitation (i.e. how to solve the given task during test after training is done); do we still use beam search?
>
> For exploitation (test time) in the Common Core environments, we only use the greedy policy induced by the score function (i.e. take the top prediction at each state, without performing search). This was specified in Section 5.2, but we see it is missing from the algorithm description! For the Rubik's Cube, we use the same variant of A* search during test time as DeepCubeA.

---

> > ### Comment · Reviewer_xzcf · 2021-09-02
> > **Response to rebuttal**
> >
> >
> > I appreciate the authors providing detailed feedback. My apologizes for the late response, because I had a difficult time reassessing the work more carefully and for other personal matters as well. My major concerns were about (i) RL setup, (ii) unstructured text, and (iii) experimental evaluation.
> >
> > #### 1. Reinforcement Learning Setup?
> >
> > It was my misunderstanding that during test time, MCTS / beam search is not used (which should be clearly described in the paper as well). I can agree that the task is a sequential decision making problem, formalized in the context of MDP with proper reward functions and agent’s policy (action space), which should be more clearly described in some background sections.
> >
> > However, the algorithm doesn’t follow quite conventional approaches in RL; as the author mentioned, there are no TD, no value learning, no direct policy search, no model learning, etc. as the policy is simply the greedification of the similarity function $f(p, s_t)$. If we interpret this as some sort of an action-value function, it might be seen as if we are doing a contrastive learning or ordinal regression of such a Q function. Can the authors provide additional discussion about this more clearly?
> >
> > #### 2. Unstructured text:
> >
> > Thank you for the clarification. Please discuss this point more clearly. The paper should be VERY careful in saying “unstructured”. Saying unstructured text for inputs to the network (i.e. observation, processed by a LSTM encoder) is fine, and I think this is great due to general applicability, and thus welcome to be emphasized, but probably not for the action space. So as to derive a next state $p$ the structure of the environment is already compromised. In particular, I think L49 has a false claim which should be toned down; it looks like action spaces aren’t unstructured.
> >
> > #### 3. Difficulty of the environment and Comparison with other baselines.
> >
> > I’ve read the response so many times, and I am more convinced about the result. Comparison with ADI looks good. I cannot get how exactly the A*/IDA were implemented, but actually the comparison results with A*/IDA (86% fraction, 6% equation, etc.) are pretty surprising. I believe they actually could work much better (for example, sorting could be ~100% given L <= 11). Why only a million nodes to visit, in the first place? CoPoLE has used 10M steps during training. Although I am not sure this is a fair comparison, I can project that even with 10M steps ConPoLe might work better because it is far better able to prioritize more probable actions.
> >
> > All that said, however, the CommonCore problems still seem contrived and not a very difficult search domain. Rubik’s cube is a more challenging and difficult domain. I think ConPoLE contributes towards a general solver, and combination with a search-based algorithm (e.g. using ConPOLE for A* heuristics or in combination with MCTS) will be also interesting to study. Given such addressed points, I am increasing my rating from 5 to 6 (slightly on a borderline).

---

> ### Author Response · Authors · 2021-08-27
> **Comments or suggestions?**
>
> We wonder if the reviewer had a chance to consider our new results, comparing against other existing baselines (especially ADI on the RL side, and MathSteps on the expert heuristic side). Hopefully these help give more perspective on the challenges of the domains we introduce. We also hope that the clarification about the test-time evaluation (that we do not run beam search at test time, instead using the greedy policy) help address the concern about the validity of the Reinforcement Learning setup. We’re happy to clarify these or other points further if the reviewer has any outstanding concerns, and would really appreciate comments or suggestions in light of the new comparisons.

---

### Official Review · Reviewer_z1yq · 2021-07-21

**Rating:** 4
**Confidence:** 4

**Summary:**

This paper presents four symbolic reasoning domains that are inspired by the common core curriculum and a policy learning scheme that uses a distance metric as a value heuristic, for planning.

**Ethical Concerns:**

No.

**Limitations And Societal Impact:**

The authors have adequately addressed the limitations and potential negative societal impact of their work

**Main Review:**

The motivation of this paper is to produce programs that can automatically solve some of the common common core curriculum problems in math. The domains are interesting, whereas the learning algorithm seems to produce a good performance. I enjoyed reading this paper quite a bit, but I do think the domains presented are on the easier side, and there are not enough baselines (like other methods), so I am going to recommend a weak reject.

As a method paper, it is important to evaluate the method on established dataset/problem domains, and compare the method against other baselines.

Pros of this paper are: the paper is largely well written. The domains are well motivated (not from the method perspective, because there are other domains from prior works, but well-motivated by the common core curriculum).

Cons: The second paragraph in the introduction is distracting and does not help with the main message of this paper. Being a good tutor does not have much to do with having a good automatic solution engine, and this paper does not show that this engine helps with the human subjects during learning. Technically, it is important not to ignore domains that prior work has used and just dive into designing your own domains, then run just your own method. Nonetheless, these exciting results are a good starting point for something cool!

**Time Spent Reviewing:**

5 hours, split over two days

---

> ### Author Response · Authors · 2021-08-10
> **Response and new comparisons**
>
> We thank the reviewer for the encouraging comment regarding the Common Core environments. We completely agree that more baselines would be valuable. As suggested by "reviewer cgrc", we ran Autodidatic Iteration and Deep Approximate Value Iteration, two other suitable baselines from the DeepCube family of papers, on the Common Core domains. We also tested A*/IDA search heuristics to give more perspective on the state size of the domains. Please refer to our General Comment for the results and insights from these comparisons. We would be pleased to add additional baselines that the reviewers suggest!
>
> > Being a good tutor does not have much to do with having a good automatic solution engine, and this paper does not show that this engine helps with the human subjects during learning.
>
> We agree that there is a gap between being able to solve domains and to teach them. Our motivation for working on learning solvers stems from the observation that many automated tutors, for example Google's Socratic app, the Cognitive Tutor Algebra, and the Algebra Notepad (three we cited), incorporate a solver component as a key part of their tutoring system. These solvers can be expensive to develop and error prone (see General Comment about the MathSteps solver). So our intended contribution is to make developing this building block easier -- which is indeed separate from the problem of how to best use it for tutoring. (As implied in your comment, the latter would certainly require human studies.) Besides the solver itself, we believe there's much potential in using the learned representations in a number of interesting ways (such as finding semantically similar exercises). We will clarify this motivation in the introduction!
>
> > As a method paper, it is important to evaluate the method on established dataset/problem domains, and compare the method against other baselines.
>
> We agree, and would like to emphasize that we're very open to performing more comparisons, either in existing related domains or with applicable baselines that we may not be aware of. We refer to the general comment where we explain why the baselines we have found in the RL for automated theorem proving literature and RL for text-based environments were unsuitable for a comparison. However, it's possible that more interesting comparisons have simply escaped our eye. We're certainly open to implementing other baselines or trying ConPoLe in other suitable domains, so any suggestions here would be greatly appreciated!

---

> ### Author Response · Authors · 2021-08-27
> **Comments or suggestions?**
>
> We wonder if reviewer z1yq had a chance to consider our update with the comparison against Autodidatic Iteration and Deep Approximate Value Iteration (recent RL methods) and MathSteps (hand-written expert solution)? We hope that including these new comparisons in the paper will address the reviewer's concerns about mostly running just our own method on the Common Core domains. We would really appreciate comments or suggestions from the reviewer of other baselines or comparisons that would improve our work further.

---

> > ### Comment · Reviewer_z1yq · 2021-09-10
> > **Thank you for these updates!**
> >
> > I have read the responses from the authors both under my review, and the other reviewers' comments. The added baselines, Autodidactive Iteration and DQVI are both good baselines, and I appreciate the reviewer's effort.
> >
> > I decided to maintain the score because the main issue is the positioning and the paper is an application on "contrived domains" quoting reviewer xzcf, a weakness that is difficult to address sans an updated draft.
> >
> > That said, I still believe that this paper has value, and like the general direction. I would love to see this paper appearing as a preprint in a more cleaned-up form.

---

> > > ### Author Response · Authors · 2021-09-13
> > > **Thank you!**
> > >
> > > Thank you for the last comments and general encouragement! If you don't mind, when you mentioned "other domains from prior works", is there any specific domain you had in mind?
> > >
> > > While several learning domains for formal mathematics have been released in the last years (e.g., Gamepad [1], IsarStep [2], and Mizar [3]), we note that they all have very different and specialized problem formulations. For instance, in GamePad, based on Coq, the main challenge is learning from human proofs. In IsarStep, the problem is not producing a full solution, but only predicting key terms in the middle of a complex proof template. In Mizar, the goal is closer to ours in learning step-by-step solutions purely from interaction, but refutational CNF proofs have a tree structure. With the Common Core environments, we hoped to unify similar domains from educationally-motivated works (e.g., Algebra Notepad and MathSteps both have a compatible formulation of step-by-step equation solving) in a common interface.
> > >
> > > We're excited to think about extensions of our method to these domains, but the extension will certainly look vastly different in each of them (e.g., with an imitation learning component in Gamepad, or with a sophisticated premise selection component in Mizar). So just to use this opportunity with reviewer z1yq to improve our work, do you have thoughts on specific domains you felt were missing from our evaluation? Or was it more on the feeling of not seeing how our method is positioned in the context of domains like the ones we mentioned above?
> > >
> > > [1] HUANG, Daniel et al. GamePad: A Learning Environment for Theorem Proving. In: International Conference on Learning Representations. 2018.
> > >
> > > [2] LI, Wenda et al. IsarStep: a Benchmark for High-level Mathematical Reasoning. In: International Conference on Learning Representations. 2021.
> > >
> > > [3] KALISZYK, Cezary et al. Reinforcement learning of theorem proving. In: Proceedings of the 32nd International Conference on Neural Information Processing Systems. 2018. p. 8836-8847.

---

### Author Response · Authors · 2021-08-10
**General response, new comparison to Autodidatic Iteration and MathSteps**

We thank the reviewers for their time and effort to provide constructive feedback. We appreciate the general encouragement regarding the setup and the Common Core environments we propose, especially the comments about the environments posing a "novel and interesting RL problem" and that the "environments introduced in this paper are valuable to the RL community".

Overall, the main concern seems to be whether the tasks we introduce are genuinely difficult and, relatedly, the primary suggestion was for comparisons with other baselines.

Regarding baselines, we agree that comparison to existing strong RL algorithms would be extremely helpful (indeed we initially planned to simply use existing approaches for these new domains, instead of developing a new one). Frustratingly, we had trouble finding applicable baselines that could be applied to our domains, since (1) the literature on RL for theorem proving tends to specialize in a single prediction problem tied to a particular theorem prover, and (2) the literature on RL for text-based games has focused on exploiting natural language pre-training and knowledge representation, which doesn't apply in the symbolic setting. Below we describe results for the two baselines suggested by reviewers. We would greatly appreciate additional suggestions on new comparisons to perform!  Yet, we have also come to believe that the difficulty of finding baseline algorithms reflects the novelty of the learning problem posed by our mathematics education domains.

We implemented Reviewer cgrc's suggestion of running Autodidatic Iteration (ADI) on the Common Core domains. We also tried Deep Approximate Value Iteration (DAVI, the newest algorithm in the DeepCube papers, which is a small variation, predicting cost-to-go instead of expected reward), but for us they performed very similarly. The maximum solving rates achieved when learning in the setup we originally tested (10^7 environment steps, evaluation of greedy policy only during test-time) were:

Equations: 5.5%
Fractions: 46.5%
Sorting: 91%
Ternary addition: 63.5%

Thus, ADI was in general a stronger baseline than DRRN, but still performed much worse than ConPoLe on most domains.
The stronger performance of ADI/DAVI on sorting is easy to interpret. In the Rubik's Cube, the problem generator is guaranteed to produce states with each possible number of steps away from the solution (1, 2, 3, …). This also happens in sorting, where our generator uniformly shuffles the input list, generating lists with any number of inversions. However, this assumption is violated in our other environments, where we don't have such control (e.g., in equations, our problems come from an existing educational dataset). While the DeepCubeA paper suggests this gap can be covered by performing a deeper BFS during value iteration, we found that to not be sample efficient in our setting - after seeing the same number of data points, agents performed worse.

We also implemented stronger heuristic search baselines, as suggested by reviewer xzcf's comment. In particular, A*/IDA with a "state length" search heuristic. Visiting up to a million nodes, this approach was able to solve 86% of the test problems in fractions, 51% in ternary addition, 76% in sorting, and only 6% in equations. While these results can certainly be improved (especially with a combination of learning as search, as DeepCube does in their environment), this evaluation is very different than the one we used for ConPoLe: during test time, we only used the top prediction from the model and a single roll-out in each problem, without performing any search. One advantage of performing well under this more strict evaluation is performance: combining considerable search with neural models tends to be slow, and we want our models to potentially be used interactively with human students.

Even manually writing robust heuristics for these domains is quite challenging. As we briefly touched on in the introduction, we also tested the Google MathSteps library, which is used by real-world educational apps as an equation solver. MathSteps implements several solving heuristics for equations (without performing search), spanning more than 7k lines of code. Surprisingly, it only succeeds at 69.3% of our test equations. (The remaining ones trigger assertions in its planning logic, such as performing moves that eliminate all occurrences of the variable, or reaching a state where it doesn't expect there to be parentheses. We debugged a few of these cases manually, and found that small modifications to the equations usually make them solvable. Thus, MathSteps has all the features it needs to solve all equations, but our tests found several unhandled corner cases in its heuristics.)

We hope these results provide more perspective on the challenge posed by our setup. We thank the reviewers for suggestions, as these new insights will strengthen our paper. Meanwhile, we're still open to trying other stronger learning baselines that any reviewer might be aware of that we might have missed!

---

### Decision · Program_Chairs · 2021-09-27

**Decision:**

Accept (Poster)

**Comment:**

I've read through the reviews and responses as well as the paper. It seems like the reviewers found the paper very interesting: the problem domain is compelling and under-explored, and the motivation of the model is clear. In other words, the reviewers generally felt the paper was well-positioned to bring value to the field of abstract reasoning and planning, and the authors did a decent job at demonstrating this through benchmarks introduced here as well as showing some reasonable baselines fall short.

However, there were some concerns, and I was borderline as far as whether these concerns are addressable through a minor revision in the final draft. It seems like a lot of the issues are on the stated limitations of the work: the authors acknowledge in the discussion some of these limitations are properties of the domain they are working in.

After looking carefully at the discussion and the paper, I believe the paper still has value, despite these limitations. We shouldn't detract good research in difficult domains, and the fact that there's more to say about this domain shouldn't make it weaker than a paper that purports to study reasoning in a black-box environment like Atari. However, I strongly assert that the authors make very clear the limitations of their benchmarks and conclusions draft from running models on these benchmarks in the final draft. I cannot conditionally accept the paper, so I am going to trust the authors on this.